Thymines opposite to bulky aristolactam-DNA adducts in duplex DNA are not targeted by human thymine-DNA glycosylase

http://orcid.org/0000-0003-3371-4459 Manapkyzy Diana 1 2
Zhamanbayeva Gulzhan 1 2
Sidorenko Viktoriya 3
Bonala Radha 3
Johnson Francis 3 4
http://orcid.org/0000-0003-0775-7324 Matkarimov Bakhyt T. 5 6
http://orcid.org/0000-0001-5013-0194 Zharkov Dmitry 7 8
Saparbaev Murat K. 2 9 murat.saparbaev@gustaveroussy.fr
Taipakova Sabira 1 2 sabira.taipakova@gmail.com
1 Scientific Research Institute of Biology and Biotechnology Problems, Al-Farabi Kazakh National University , Almaty , Kazakhstan
2 Department of Molecular Biology and Genetics, Faculty of Biology and Biotechnology, Al-Farabi Kazakh National University , Almaty , Kazakhstan
3 Department of Pharmacological Sciences, State University of New York at Stony Brook , Stony Brook, NY , United States
4 Department of Chemistry, State University of New York at Stony Brook , Stony Brook, NY , United States
5 L. N. Gumilev Eurasian National University , Astana , Kazakhstan
6 National Laboratory Astana, Nazarbayev University , Astana , Kazakhstan
7 Department of Natural Sciences, Novosibirsk State University , Novosibirsk , Russia
8 Institute of Chemical Biology and Fundamental Medicine, Siberian Branch of the Russian Academy of Sciences , Novosibirsk , Russia
9 Gustave Roussy Cancer Campus—UMR9019 CNRS, Université Paris-Saclay , Villejuif , France
Uversky Vladimir
Electronic publication date: 2025 Jul 4
Publication date: 2025
Volume: 13
Electronic Location ID: e19577
Received 2025 Feb 13; Accepted 2025 May 19
Copyright: © 2025 Manapkyzy et al.
Copyright year: 2025
Copyright holder: Manapkyzy et al.
License: This is an open access article distributed under the terms of the Creative Commons Attribution License, which permits unrestricted use, distribution, reproduction and adaptation in any medium and for any purpose provided that it is properly attributed. For attribution, the original author(s), title, publication source (PeerJ) and either DOI or URL of the article must be cited.
License URL: https://creativecommons.org/licenses/by/4.0/

Keywords: DNA glycosylase, DNA repair, Base excision repair, Aristolochic acid, Aristolactam-adenine adduct, T to A transversions, Mutational signature, Thymine-DNA glycosylase, Aberrant DNA repair, Aristolactam-guanine adduct

Funding: Committee of Science of the Ministry of Science and Higher Education of the Republic of Kazakhstan AP19676334, BR24992841, BR24993023 Nazarbayev University CRP 111024CRP2009 French National Research Agency ANR-22-CE12-0034-01 Electricité de France RB 2021-05 Russian Ministry of Higher Education and Science FSUS-2020-0035 This work was supported by grants from the Committee of Science of the Ministry of Science and Higher Education of the Republic of Kazakhstan grant AP19676334 to Sabira Taipakova, and grants BR24992841 and BR24993023 and Nazarbayev University CRP grant 111024CRP2009 to Bakhyt T. Matkarimov; French National Research Agency (ANR-22-CE12-0034-01) and Electricité de France RB 2021-05 to Murat K. Saparbaev; Russian Ministry of Higher Education and Science (FSUS-2020-0035) to Dmitry Zharkov; Diana Manapkyzy and Gulzhan Zhamanbayeva were supported by fellowships Abai-Vern and Bolashaq, Kazakhstan, respectively. The funders had no role in study design, data collection and analysis, decision to publish, or preparation of the manuscript.

==============================
Background

Consumption of aristolochic acids (AA) from the plant Aristolochia results in the formation of bulky aristolactam-dA (dA-AL) and aristolactam-dG (dG-AL) adducts in cellular DNA ultimately leading to the development of urothelial cancer. Intriguingly, the dA-AL adducts induce A•T→T•A transversions in tumor cells preferentially in CpA*→TpG context. The human mismatch-specific thymine-DNA glycosylase (TDG) protects cells against mutagenesis induced by spontaneous deamination of 5-methylcytosine (5mC) by removing thymine opposite to guanine in a CpG context in the base excision repair (BER) pathway. Nevertheless, challenges for DNA glycosylases to the faithful discrimination between non-damaged and damaged DNA strands do exist, such as mismatched pairs between two canonical bases, which may result due to DNA polymerase errors during replication. Previously, we demonstrated that TDG is prone to aberrant excision of T opposite to damaged adenine in duplex DNA in CpA*/TpG context.

Methods

In the present work, using in vitro reconstitution assays, we investigated whether TDG participates in the aberrant removal of thymine opposite to dA-AL adducts in duplex DNA.

Results

We have demonstrated that TDG either does not excise thymine or does so with extremely low efficiency when it is paired with dA-AL or dG-ALII adducts in duplex DNA. At the same time, TDG excises with high efficiency thymine opposite to guanine and hypoxanthine in T•G and T•Hx mispairs.

Discussion

These findings strongly suggest that the human TDG is not involved in the aberrant DNA repair of AA-induced DNA damage.

Introduction

To counter mutagenic threats to genome stability caused by endogenous and environmental factors such as spontaneous DNA decay and oxidative stress, cells evolved several DNA repair pathways. One of these, base excision repair (BER), employs a number of DNA glycosylases, the enzymes that recognize irregular bases and excise them from DNA, producing an apurinic/apyrimidinic (AP) site, which is then cleaved by AP endonuclease (Hitomi, Iwai & Tainer, 2007; Krokan & Bjoras, 2013) (Fig. 1A). In addition to processing damaged bases, some DNA glycosylases are mismatch-specific, recognizing and removing regular DNA bases from non-canonical base pairs. In contrast to the replication-dependent mismatch repair pathway dedicated to DNA polymerase errors correction, mismatch-specific DNA glycosylases do not discriminate between newly synthesized and parental DNA strands and thus can potentially cause mutations if they remove a base originally present in the genome.

Figure 1 Schematic representation of the canonical and aberrant base excision repair (BER) pathways.

(A) The faithful, canonical BER pathway take in charge G•T mispair, resulting from spontaneous deamination of 5mC (Cm) in CmpG/CpG context (1). Human TDG excises T opposite to G (2) and initiates series of steps (3–4) that restore DNA primary structure. Note that if G•T mispair is not repaired, DNA replication (denoted by blue arrow) would lead to CmpG/CpG → TpG/CpA mutation. (B) The aberrant, mutagenic BER pathway removes thymine opposite to hypoxanthine (Hx) in Hx•T base pair, which arises due to spontaneous deamination of adenine in CpA/TpG context (1). Human TDG excises T opposite to Hx in aberrant manner (2) and initiates series of steps (3–4) that introduce CpA/TpG → CpG/CpG mutation via DNA replication (denoted by blue arrow).

Post-replicative DNA methylation, the enzymatic transfer of a methyl group onto the fifth carbon of cytosine (C) in the CpG context, plays a crucial role in the embryonic development, cell differentiation, suppression of mobile genetic elements and genomic imprinting (Bird, 2002; Greenberg & Bourc’his, 2019; Smith & Meissner, 2013). One disadvantage of DNA methylation is the hypermutability of 5-methylcytosine (5mC) because of spontaneous deamination, which yields thymine (T) and gives rise to a thymine to guanine (G) mismatch. In the absence of repair, the mispaired thymine leads to a C→T transition after replication. In mammals, two mismatch-specific DNA glycosylases, methyl-binding domain protein 4 (MBD4/MED1) and thymine-DNA glycosylase (TDG), offer protection against this mutagenic effect by removing the mismatched T, which is then replaced with regular C in the canonical BER pathway (Hendrich et al., 1999; Neddermann & Jiricny, 1993) (Fig. 1A). TDG, however, displays much wider substrate specificity: its main substrates are products of targeted 5mC oxidation by TET dioxygenases in the process of active epigenetic demethylation (Cortazar et al., 2011; Cortellino et al., 2011), and it can also excise mismatched uracil (Neddermann & Jiricny, 1994), its modifications at C5 (Hardeland et al., 2003), 3,N4-ethenocytosine (Hang et al., 1998; Saparbaev & Laval, 1998), 5-hydroxycytosine (Bennett et al., 2006), thymine glycol (Yoon et al., 2003), and 7,8-dihydro-8-oxoadenine (8oxoA) (Talhaoui et al., 2013). Earlier, we showed that MBD4 and TDG catalyze aberrant excision of regular T (Talhaoui et al., 2014): in damaged DNA, TDG can target the normal strand and excise T opposite 1,N6-ethenoadenine (εA), hypoxanthine (Hx), 8oxoA and the AP site in the TpG/CpX sequence context (where X is a damaged residue). MBD4 can excise T opposite εA but not opposite Hx or other A modifications. Reconstitution of TDG-initiated BER of T•Hx mispairs in vitro shows that DNA polymerase β uses the damaged strand as a template resulting in a T→C mutation even in the absence of replication (Talhaoui et al., 2014) (Fig. 1B). Thus, the same DNA repair systems that counteract genotoxic effects of DNA damage, could act in aberrant ways, which may contribute to cancer and neurodegenerative diseases.

Here and below, to discriminate between different forms of non-canonical DNA repair activities, which may take place under specific conditions, we use terms “futile” and “aberrant”. Briefly, futile activity is the action of a repair enzyme on fully normal DNA, which initiates an idle round of nucleotide removal and replacement, as in the excision of normal purines by alkylpurine–DNA glycosylases (Berdal, Johansen & Seeberg, 1998) or the excision of undamaged oligonucleotide fragments by nucleotide excision repair (NER) machinery (Branum, Reardon & Sancar, 2001). Aberrant repair, on the other hand, is the repair of damaged DNA, that promote a mutation rather than restoration of the original sequence. As examples, one may consider the action of MutY when it removes A opposite a misincorporated 7,8-dihydro-8-oxoguanine (8oxoG) residue and thus may trigger an A→C transversion (Fowler et al., 2003), or removes A from A:G and A:C mismatches in a manner indiscriminate of the parent/daughter strand status (Lu & Chang, 1988; Radicella, Clark & Fox, 1988; Su et al., 1988) or the action of human MMR when it recognizes T preferentially misincorporated opposite to O6-methylguanine initiating multiple rounds of strand degradation and resynthesis ultimately leading to chromosomal instability (Hampson et al., 1997). Nevertheless, it should be stressed that bacterial MutY and human MMR play predominantly anti-mutagenic role, since in their absence cells exhibit increased spontaneous mutation rates (Hirano et al., 2003; Modrich & Lahue, 1996).

Environmental carcinogens present an important source of genomic instability. In particular, consumption of plant products containing aristolochic acid (AA) as alternative medicines has been connected with high risk of upper urinary tract cancer and permanent kidney damage. Once in the body, AA is transformed by cellular nitroreductases to highly reactive intermediates that covalently bind to purines in DNA to yield 7-(deoxyadenosine-N6-yl)aristolactam (dA-ALI and dA-ALII) (Figs. 2A, 2B) and 7-(deoxyguanosine-N2-yl) aristolactam I (dG-AL) adducts (Figs. 2C, 2D) (Bieler et al., 1997). In the resulting tumors, the dA-AL adducts generate a unique mutational pattern characterized by A→T transversions positioned on the non-transcribed DNA strand (Moriya et al., 2011). Remarkably, dA-AL adducts are not recognized by global genome NER, but can be efficiently removed by transcription-coupled NER, thus explaining their highly mutagenic properties and the strand bias (Sidorenko et al., 2012). The resistance to global genome NER might be due to the peculiar conformation of dA-AL adducts, which, as revealed by nuclear magnetic resonance spectroscopy, can stack between the flanking bases without strongly perturbing the DNA duplex (Lukin et al., 2012). In the absence of repair, DNA polymerase ζ is predominantly responsible for both error-free and mutagenic bypass of bulky dA-ALI, incorporating both dAMP and dTMP across from it (Hashimoto et al., 2016).

Figure 2 Chemical structures of aristolactam (AL)-DNA adducts.

(A) dA-ALI; (B) dA-ALII; (C) dG-ALI; (D) dG-ALII generated via bioactivation of aristolochic acids in human cells.

Whole-exome sequences of upper urinary tract carcinoma (UTUC) samples of patients from Taiwan, the region with the highest AA consumption in the world, revealed two types of tumors with distinct mutational signatures: one resembling that recorded in UTUCs elsewhere, another showing a greatly elevated number of mutations mainly A→T transversions (Hoang et al., 2013). Strikingly, the most prominent hot spot of the AA-related mutations was in a CpTpG context (CpTpG→CpApG or CpA*pG→CpTpG on the complementary strand, where A* is a damaged adenine) (Ng et al., 2017). This mutational signature closely resembles that expected from TDG-catalyzed aberrant excision of T opposite to damaged A in its preferred sequence context (TpG/CpX, where X is a modified adenine), as described in our previous study (Talhaoui et al., 2014). Based on these observations, we suggested that TDG may participate in the aberrant excision of T opposite to dA-AL adducts, thus explaining the observed mutational signature.

One more feature of AL adducts had drawn particular attention to TDG as a possible agent of mutagenesis. The NMR structure of dA-ALII (Lukin et al., 2012) and molecular dynamics models of dA-ALI, dA-ALII and dG-ALII (Kathuria et al., 2015; Kathuria, Sharma & Wetmore, 2015; Kathuria, Sharma & Wetmore, 2016) show that the extended planar system of AL neatly intercalates in the DNA stack pushing the opposite base out of the helix. This tendency is more pronounced for dA-AL•T than for dG-AL•C pairs (Kathuria, Sharma & Wetmore, 2015). In a structurally similar situation, placing a bulky synthetic pyrene nucleotide opposite to U or even C rescues a mutant of uracil–DNA glycosylase (UNG) lacking a critical Leu wedge residue used to stabilize the damaged nucleotide in the enzyme’s active site (Jiang, Kwon & Stivers, 2001; Jiang, Stivers & Song, 2002; Kwon, Jiang & Stivers, 2003). In a proposed mechanism of initial lesion recognition by UNG, the enzyme captures a spontaneously open uracil-containing pair and uses this wedge to prevent the damaged base from falling back into the stack (Cao et al., 2004; Krosky, Song & Stivers, 2005; Parker et al., 2007). Since TDG is a homolog of UNG, we wondered whether forced eversion of a normal pyrimidine nucleotide by the AL moiety might promote the aberrant activity of the enzyme.

In the present study, we examined TDG activities toward damaged DNA duplexes containing aristolactam-DNA adducts (aberrant activity) and non-damaged regular DNA duplexes (futile activity). We demonstrated that during long incubation at 37 °C TDG can excise, albeit with low efficiency, both regular pyrimidine residues in non-damaged DNA and thymine residues opposite to dA-ALI, dA-ALII or dG-ALII adducts in damaged DNA duplexes.

Materials and Methods

Proteins

Phage T4 polynucleotide kinase was purchased from New England Biolabs (Evry, France). The E. coli BL21 Arctic Express (DE3) cells were purchased from Novagen-EMD Biosciences (Merck Chemicals, Nottingham, UK). The purified human uracil-DNA glycosylase (hUNGΔ84) was from laboratory stock (Morera et al., 2012).

The native full-length TDG1–410 (TDGFL) protein was produced as described previously (Talhaoui et al., 2013). In short, Arctic Express (DE3) cells were transfected with the expression vector pET28c-TDGFL (for full-length TDG protein) and the transformants were grown on a shaker at 37 °C in LB, accompanied with 50 µg/ml of kanamycin, to OD600 nm = 0.6–0.8. Thereafter, the temperature was decreased to 12 °C, the recombinant protein expression was induced by 0.2 mM isopropyl β-D-1-thiogalactopyranoside and the cultures were grown for another 15 h. The cells were collected by centrifugation and the bacterial pellets were lysed by a French press at 18,000 psi in buffer composed of 40 mM NaCl, 20 mM HEPES–KOH (pH 7.6) and 0.025% Nonidet P-40, complemented with Complete™ Protease Inhibitor Cocktail (Roche Diagnostics, Switzerland). The supernatants were collected after centrifugation of cell lysates at 40,000× g for 1 h at 4 °C, and adjusted to 20 mM imidazole and 500 mM NaCl before loading onto a HiTrap Chelating HP column (Amersham Biosciences, GE Healthcare, Chicago, IL, USA). All purification procedures were done at 4 °C. The column was rinsed with Buffer A1 (500 mM NaCl, 20 mM imidazole and 20 mM HEPES), and the remaining proteins were eluted with a continuous gradient of 20–500 mM imidazole in Buffer B1 (20 mM HEPES–KOH pH 7.6, 500 mM NaCl, 500 mM imidazole). The proteins were further purified by employing a Heparin HP affinity column. The fractions after HiTrap Chelating HP were collected and diluted tenfold in Buffer A2 (50 mM NaCl, 20 mM HEPES pH 7.6 and 0.0125% Nonidet P-40) and loaded on a 1-ml HiTrap Heparin column (Amersham Biosciences, Orsay, France). Proteins bound to the Heparin column were eluted by a 50–800 mM NaCl gradient in Buffer B2 (1,000 mM NaCl, 20 mM HEPES–KOH pH 7.6 and 0.0125% Nonidet P-40). Column fractions were analyzed by SDS-PAGE and the fractions with the pure His-tagged TDGFL protein were kept at −80 °C in 50% glycerol. Concentrations of recombinant proteins in column fractions were measured by the Bradford assay. The activities of purified proteins were verified by using their classical DNA substrates immediately prior to use.

Oligonucleotides

The DNA oligonucleotide sequences used in this study are shown in Table 1. The oligonucleotides containing dA-ALI, dA-ALII and dG-ALII adducts were synthesized as described previously (Attaluri et al., 2010, 2014). All of the oligodeoxyribonucleotides composed of regular DNA base residues including their complementary counterparts and also oligonucleotides containing modified bases such as hypoxanthine (Hx) and uracil (U) were purchased from Eurogentec (Seraing, Belgium). The oligonucleotides were 5′-end labelled with γ-[32P]-ATP (PerkinElmer, Waltham, MA, USA) and then annealed with corresponding complementary strands as described previously (Gelin et al., 2010). The ensuing duplex oligonucleotides are denoted to as X*•N, where X is a modified or regular nucleobase in the 32P-labelled strand and N is a canonical DNA base opposite to X in the non-labelled complementary strand.

Table 1 DNA sequence of the oligonucleotides used in the study.

ID	Length	Sequence (5′→3′)	Comment	Reference	
14-03	27	CCATCATCTCCAGACXGATCCTCACAC	Contains G, Hx, dA-ALI or A (X) at position 16 in the CpXpG context.	Hashimoto et al. (2016)	
c14-03	27	GTGTGAGGATCTGTCTGGAGATGATGG	Complementary to 14-03, places T opposite to X in 14-03.		
c1403-55	55	GGTACGTTCGTACCGTGTGAGGATCTGTCTGGAGATGATGGGCCGTCTTGACGGC	Forms 6-bp hairpins at both ends exposing a 27-nt gap (underlined) complementary to 14-03, places T opposite to X in 14-03.	Bazlekowa-Karaban et al. (2019)	
14-03-U11	27	CCATCATCTCUAGACAGATCCTCACAC	Contains U at position 11 in the CpUpA context.		
14-04	27	CCATCATCTCCAGAAXTATCCTCACAC	Contains G, Hx, dA-ALI or A (X) at position 16 in а ApXpT context, otherwise identical to 14-03.	Hashimoto et al. (2016)	
c14-04	27	GTGTGAGGATATTTCTGGAGATGATGG	Complementary to 14-04, places T opposite to X in 14-04.		
c1404-55	55	GGTACGTTCGTACCGTGTGAGGATATTTCTGGAGATGATGGGCCGTCTTGACGGC	Forms 6-bp hairpins at both ends, exposing a 27-nt gap (underlined) complementary to 14-04, places T opposite to X in 14-04.	Bazlekowa-Karaban et al. (2019)	
14-06	19	TTCCCTCCAGAAXCATCCT	Contains dA-ALI or A (X) at position 13 in а ApXpC context.	Attaluri et al. (2010)	
c14-06	19	AGGATGTTTCTGGAGGGAA	Complementary to 14-06, places T opposite to X in 14-06.		
14-07	24	TCTTCTTCTGTGCXCTCTTCTTCT	Contains dA-ALI or A (X) at position 14 in а CpXpC context.	Attaluri et al. (2010)	
c14-07	24	AGAAGAAGAGTGCACAGAAGAAGA	Complementary to 14-07, places T opposite to X in 14-07		
10-05	24	TCTTCTTCTGTGCYCTCTTCTTCT	Contains dA-ALII or A (Y) at position 14 in a CpYpC context.	Attaluri et al. (2010)	
c10-05	24	AGAAGAAGAGTGCACAGAAGAAGA	Complementary to 10-05, places T opposite to Y in 10-05.		
10-13	24	TCTTCTTCTZTGCACTCTTCTTCT	Same as 10-05 but contains dG-ALII or G (Z) at position 10 in a TpZpA context.	Attaluri et al. (2010)	
c10-13	24	AGAAGAAGAGTGCATAGAAGAAGA	Complementary to 10-13, places T opposite to Z in 10-13.		

DNA repair activity assays

Unless otherwise specified, the standard reaction mixture (20 µl) contained 20 nM 32P-labelled duplex oligonucleotide, 20 mM Tris–HCl (pH 8.0), 1 mM ethylenediamine tetraacetate (EDTA), 100 µg/ml bovine serum albumin (BSA), 1 mM dithiothreitol (DTT) and 200 nM TDGFL. The reaction mixes were incubated at 37 °C for 5–60 min when assessing specific TDG-activities (G•T* and G•U* duplexes; the asterisk indicates the labelled strand having the defined base in a specific position) or 1, 6 and 18 h when measuring TDG-catalyzed aberrant or futile activity (T*•dA-ALI, T•dA-ALI*, T*•dA-ALII, T•dA-ALII*, T*•dG-ALII, T•dG-ALII*, T*•A, T•A* duplexes). The reaction mix for UNGΔ84 composed of 20 mM Tris–HCl (pH 8.0), 100 mM NaCl, 1 mM DTT, 100 µg/ml BSA, 1 mM EDTA was incubated at 37 °C for 30 min. All reactions were arrested with 10 µl of a stop solution consisting of 20 mM EDTA and 0.5% SDS. Unless otherwise specified, after incubation, the samples were incubated with 0.1 M NaOH for 3 min at 95 °C to cleave AP sites remained after DNA glycosylases catalyzed base excision, following the mixes were neutralized with HCl. Alternatively, the samples were treated with light piperidine (LP): 10% piperidine for 30 min at 37 °C and then neutralized with HCl. The reaction mixture for examining the combined AP endonuclease and 3′→5′ exonuclease activity of APE1 contained 200 nM TDGFL in 20 mM Tris–HCl (pH 8.0), 1 mM DTT, 100 µg/ml BSA, 1 mM MgCl2 and was incubated for 1 h at 37 °C. Subsequently increasing concentrations of APE1 (5–200 nM) were added to the sample and the incubation was continued for another 3 h. All samples were desalted with a Sephadex G-25 DNA grade SF column (GE Healthcare, Chicago, IL, USA) equilibrated in 7.5 M urea and run through denaturing 20% (w/v) polyacrylamide gel (7 M urea, 0.5×TBE, 42 °C) electrophoresis. The gels were scanned using a Fuji FLA-3000 Phosphor Screen (Fujifilm, Tokyo, Japan) and Typhoon FLA 9500 imager (GE Healthcare, Chicago, IL, USA) and quantified using Image Gauge v4.0 software (Fujifilm, Tokyo, Japan). At least three independent experiments were performed for all measurements. To generate size markers, 10 nM 32P-labelled T*•G and T*•Hx duplexes were incubated with 200 nM TDGFL for 5–60 min at 37 °C.

To calculate cleavage efficiency, we employed adjusted cleaved fraction (ACF) method which is useful when short DNA cleavage fragments can be generated from long DNA cleavage fragments in DNA substrates containing multiple cleavage site. We use following formula to calculate ACF:

ACF=(TotalIntensityofCleavedBands)/TotalDNAIntensity

Total intensity of cleaved bands includes the sum of the intensities of both long and short cleaved fragments. Total DNA Intensity is the sum of intensities of the non-cleaved full-length DNA band and all cleaved fragments (long and short). The statistical analysis of quantitative data presented in Figs. 3–7 can be found in Supplemental Information (Tables S1–S5).

Figure 3 The action of TDGFL on dumbbell DNA duplexes containing dA-ALI.

TDGFL (20 or 200 nM) was incubated with 20 nM dumbbell-shaped duplex (dmbDNA) containing G, Hx or dA-ALI opposite to T at position 26 in the CpA* and ApA* sequence contexts at 37 °C for 1, 6 and 18 h. (A) Denaturing PAGE analysis of the products of 14-03 dmbDNA cleavage. Lane 1, control G•T* dmbDNA; lane 2, same as 1 but with TDGFL ; lane 3, Hx•T* dmbDNA with TDGFL ; lanes 4–5, same as 2–3, but with 100 mM NaCl; lane 6, control dA-ALI•T* dmbDNA containing an dA-ALI•T base pair; lanes 7–9, same as 6, but incubated with TDGFL for 1, 6 and 18 h, respectively; lanes 10–12, same as 7–9, but with 100 mM NaCl. Arrows on the both sides of the panel indicate the size of the DNA substrate and cleavage fragments. The small numbers below the lane numbers correspond to the percentage of the cleavage fragments, the size of fragments is denoted by red numbers. (C) Denaturing PAGE analysis of the products of 14-04 dmbDNA cleavage. Lanes 1–12, same as in (A) but with 14-04 dmbDNA. (B, D) Scheme of 41-mer 14-03 (B) and 14-04 (D) dmbDNA with the arrows marking the pyrimidines excised by TDGFL. See Table S1 for details.

Figure 4 The action of TDGFL on different strands of dA-ALI•T* dmbDNA.

TDGFL (20 or 200 nM) was incubated with 20 nM 41-mer 14-03 and 14-04 dmbDNA containing either G or dA-ALI opposite to T at position 26 in CpA and ApA contexts, respectively. (A, C) Denaturing PAGE analysis of the reaction products. Lanes 1–5 and 9–13, 41-mer dmbDNA in which the bottom 55-mer hairpin strand is 32P-labelled; lanes 6–8 and 14–16, 41-mer dmbDNA in which the top 27-mer strand with dA-ALI is 32P-labelled. Lanes 1, 3, 6, 9, 11 and 14, no enzyme; lanes 2, 4, 7, 10, 12 and 15, TDGFL for 1 h at 37 °C; lanes 5, 8, 13 and 16, TDGFL for 18 h at 37 °C. Arrows on the both sides and within the panel indicate the size of the DNA substrate and cleavage fragments. The small numbers below the lane numbers correspond to the percentage of the cleavage fragments, the size of fragments is denoted by red numbers. (B, D) Scheme of dmbDNA 14-03 and 14-04 with the arrows marking the pyrimidines excised by TDGFL. See Table S2 for details.

Figure 5 Removal of bulky dA-ALI adduct in the top 27-mer strand of dA-ALI*•T dmbDNA through combined action of TDGFL and APE1.

TDGFL (200 nM) was incubated for 1 h in a buffer containing 1 mM MgCl2 with 20 nM 41-mer A*•T or dA-ALI*•T dmbDNA containing either A or dA-ALI, respectively, in the ApA context at position 16 of the 32P-labelled 27-mer 14-04 oligonucleotide. Then both samples were either treated with light piperidine, or various amounts of APE1 (5, 20, 50 and 200 nM) were added and incubated for 3 h at 37 °C to cleave AP sites generated by TDG. (A) Reaction products separation with denaturing PAGE. Lanes 1–2, 41 mer G • T17* dmbDNA containing T • G mismatch at position 17; lanes 3–8, 41-mer A* •T dmbDNA in which the top 27-mer strand is 5′-32P labelled and contains A at position 16; lanes 9–14, same as 3–8, but dA-ALI* •T dmbDNA with dA-ALI at position 16; lanes 15–16, 27 mer 14-04 ssDNA containing Uracil at position 17. Note that G • T17* dmbDNA and U17 ssDNA were used to generate 16 mer size marker. Note that T residue 3′-next to A and dA-ALI at position 17 in the 27-mer strand is excised by TDG to generate a 16-mer product. Arrows on the both sides of the panel indicate the size of DNA substrate and cleavage fragments generated by combined action of TDG and APE1. The small numbers below the lane numbers correspond to the percentage of the cleavage fragments, the size of fragments is denoted by a red number. (B) Scheme of 41 mer dmbDNA with the arrow marking the thymine excised by TDGFL. (C) Scheme of the futile BER pathway in which APE1 removes 3′-terminal dA-ALI by its 3′→5′ exonuclease activity. See Table S3 for details.

Figure 6 The action of TDGFL on short blunt-end duplex oligonucleotides containing A, dA-ALI or dA-ALII.

TDGFL (20 or 200 nM) was incubated with 20 nM 24-mer 10-05 or 14-07 duplexes in which either the top strand with A or dA-AL at position 14 or the bottom strand with T at position 11 was 32P labelled. (A) Denaturing PAGE analysis of the products of 10-05 cleavage. Lanes 1–2, G•T11* duplex; lanes 3–5, regular A14•T11* duplex; lanes 6–8, regular A14*•T11 duplex; lanes 9–10, G•U15* duplex; lanes 11–13, damaged dA-ALII14•T11* duplex; lanes 14–16, damaged dA-ALII4*•T11 duplex. (C) Denaturing PAGE analysis of the products of 14-07 cleavage. Lanes 1–11 are the same as lanes 1–8 and 11–16 in (A) but with 14-07. Arrows on the both sides and within the panel indicate the size of the DNA substrate and cleavage fragments generated by TDG. The small numbers below the lane numbers correspond to the percentage of the cleavage fragments, the size of fragments is denoted by red numbers. (B, D) Scheme of 10-05 and 14-07 duplexes with the arrows marking the pyrimidines excised by TDGFL. See Table S4 for details.

Figure 7 The action of TDGFL on short blunt-end duplex oligonucleotides containing A, dA-ALI or dG-ALII.

TDGFL (20 or 200 nM) was incubated with 20 nM 14-06 or 10-13 oligonucleotide duplexes in which either the top strand or the bottom strand with T opposite to the adduct was 32P-labelled. (A) Denaturing PAGE analysis of the products of 14-06 cleavage. Lanes 1–2, G13•T7* duplex; lanes 3–5, regular A•T* duplex; lanes 6–8, regular A*•T duplex; lanes 9–11, damaged dA-ALI•T* duplex; lanes 12–14, damaged dA-ALI*•T duplex; lanes 15–16, G9•T11* duplex. Arrows on the both sides and within the panel indicate the size of the DNA substrate and cleavage fragments generated by TDG. The small numbers below the lane numbers correspond to the percentage of the cleavage fragments, the size of fragments is denoted by red numbers. (C) Denaturing PAGE analysis of the products of 10-13 cleavage. Lanes 1–16, are the same as lanes 1–16 in (A) but with 10-13. (B, D) Scheme of 14-06 and 10-13 duplexes with the arrows marking the pyrimidines excised by TDGFL. See Table S5 for details.

Results

Upon long incubation, human native TDG excises T opposite to A, and to a lesser extent T opposite to dA-ALI adduct in duplex DNA

Here, we were interested in studying the phenomenon of aberrant repair catalyzed by TDG on the DNA duplexes containing dA-AL adducts. To examine whether TDG could excise T opposite to dA-AL adducts in duplex DNA, we incubated the TDGFL protein with 5′-32P-labelled duplex oligonucleotides containing G, Hx and the dA-ALI adduct in two different sequence contexts: CpXpG, “favorable” for the mutagenesis by AA, referred to as 14-03 and “unfavorable” ApXpT referred to as 14-04 (were X is for G, Hx and dA-ALI). Other than this difference, the oligonucleotides were identical. In addition, to avoid non-specific degradation of the DNA substrate by possible contaminating nuclease activities, we assembled a dumbbell-shaped duplex (dmbDNA) consisting of a short 27-mer oligonucleotide containing either G or Hx or dA-ALI adduct at position 16 in 14-03 or 14-04 contexts (Fig. 3 and Table 1) and a long 55-mer strand forming hairpins at both ends and single-strand gap complementary to the 27-mer oligonucleotide (referred as “c1403-55” and “c1404-55” Table 1). The same dmbDNA construct was used in our previous study to measure activities of human major AP endonuclease 1, APE1 (Bazlekowa-Karaban et al., 2019). The 27-mer 14-03 and 14-04 oligonucleotides, containing G, Hx or dA-ALI residues were annealed to the 5′-32P labelled 55-mers c1403-55 and c1404-55, respectively, so to place a regular T opposite to the position 16. This hybridization resulted in the formation of 41-mer dmbDNA substrates in which the T26 residue is located opposite to G, Hx or dA-ALI in either 14-03 or 14-04 sequence context (Figs. 3B, 3D). Next, the dmbDNA substrates G•T*, Hx•T* and dA-ALI•T* (asterisk indicates the labelled DNA strand with the defined residue) were incubated in the presence of 20 or 200 nM TDGFL in a reaction buffer, with or without 100 mM NaCl either for 1 h or for 18 h at 37 °C, followed by the treatment with a hot alkaline to cleave AP sites produced by DNA glycosylase action. As expected, when using the DNA substrate (14-03) containing mismatched T in a favored TpG/CpX context, 20 nM TDGFL excised T opposite to G and Hx in 41-mer G•T* and Hx•T* 14-03 dmbDNA, respectively, and generated a 25-mer cleavage fragment with good efficiency, namely 6–11% of T in 1 h (Fig. 3A, lanes 2–5). TDGFL excised T opposite to G or Hx with similar efficiency and the presence of NaCl slightly inhibited the DNA glycosylase activity. In agreement with previous observations, when presented with a mismatch in an unfavorable TpT/ApX (14-04) sequence context, TDGFL excised T opposite to G or Hx with 2- or 5-fold lower efficiency, respectively, as compared to the favorable TpG/CpX (14-03) context (Fig. 3C, lanes 2–5). Interestingly, increased ionic strength markedly inhibited TDG-catalyzed aberrant excision of T opposite to Hx in an unsuitable sequence context (Fig. 3C, lane 5 vs Fig. 3A, lane 5).

In contrast, incubation of 200 nM TDGFL with 41-mer dA-ALI•T* 14-03 dmbDNA for 1, 6 or 18 h at 37 °C resulted in the generation of multiple cleavage products, the most prominent being the 35-, 29-, 25-, 17- and 15-mer cleavage fragments of variable intensities (Fig. 3A, lanes 7–9 and 10–12), indicating excision of regular T residues opposite to A at positions 36, 30, 18 and 16 in the 55-mer hairpin oligonucleotide (Fig. 3B). The TDGFL-catalyzed futile excision of T opposite to A increased in a time-dependent manner (Figs. 3A, 3C, lanes 7–12). Notably, TDGFL preferentially excised T opposite to A in position 30 of the 55-mer, cleaving 30.4% and 13.2% of T after 18 h of incubation (without or with 100 mM NaCl, respectively; Fig. 3A, lanes 9 and 12), whereas T opposite to the bulky dA-ALI adduct in the favorable TpG/CpX (14-03) context was excised at only 3.2% (no NaCl) and 1.5% (100 mM NaCl), even worse than the excision of T opposite to regular A in positions 36, 18 and 16 (lanes 9 and 12). Excision of T opposite to the bulky dA-ALI adduct in an unfavorable TpT/ApX (14-04) context was further decreased two-fold as compared to the favorable 14-03 context (Fig. 3C, lanes 9–11 vs Fig. 2A, lanes 9–11). In summary, these results suggest that the presence of dA-ALI adducts in DNA duplex: (i) does not stimulate aberrant removal of T opposite to the bulky lesion; (ii) does not interfere with futile excision of neighboring pyrimidines in the sequence contexts used.

Bulky dA-ALI adduct can be removed by APE1 3′→5′ exonuclease after excision of thymine 3′-next to the lesion by TDG

Next, we examined whether TDGFL acts on pyrimidines in the top DNA strand containing dA-ALI adduct. For this, we used dmbDNA with the 5′-32P-labelled 27-mer strand containing either G or dA-ALI in position 16 annealed to the non-labelled 55-mer hairpin oligonucleotide. As shown in Fig. 4A, TDGFL acts very weakly on the 27-mer 14-03 strand excising 1.9% of C at position 11 in CpA context after 18 h of incubation (lane 8). Strikingly, TDGFL acted more efficiently towards the 27-mer 14-04 strand, generating a 16-mer cleavage product via excision of 23.6% of T at position 17 immediately 3′ to the dA-ALI adduct (Fig. 4C, lane 16). Excision of T by TDG would generate an AP site, which then should be cleaved by APE1 to generate a single-strand break 3′ to the damaged adenine residue. This break might become persistent, since the presence of the bulky adduct at the 3′ terminus would make it a poor substrate for DNA polymerase β in the downstream steps of the BER pathway (Fig. 1). Previously, we showed that APE1 can remove bulky 8,5′-cyclopurine DNA adducts at 3′-termini of DNA strand breaks or gaps (Mazouzi et al., 2013). Therefore, we examined whether after the futile excision of T 3′-next to dA-ALI by TDG, APE1 could cleave the resulting AP site and remove the bulky adduct by its 3′→5′ exonuclease activity. For this, we incubated TDGFL with A*•T or dA-ALI*•T 14-04 dmbDNA in a buffer containing MgCl2 for 1 h and either treated with light piperidine or incubated in the presence of varying concentrations of APE1. As shown in Fig. 5A, TDGFL excised T next to A and dA-ALI with low but detectable efficiency and generated cleavage fragments after light piperidine (LP) treatment (lanes 4 and 10). When the products of reaction were incubated in the presence of 5 nM APE1, we observed a 16-mer fragment, indicating cleavage of the AP site by this enzyme (Fig. 5A, lanes 5 and 11 and Fig. 5C). Note that APE1-generated cleavage products that contain hydroxyl group at 3′-termini (lanes 5–7 and 11–13) migrated faster as compared to LP-generated fragments (lanes 4 and 10) that contain 3′-α,β-unsaturated aldehyde. Upon increasing the concentrations of APE1 up to 20 and 50 nM, we observed the appearance of the products of 3′-exonuclease degradation in both A*•T and dA-ALI*•T dmbDNA, indicative of the removal of 3′-terminal nucleotides dA and dA-ALI (Fig. 5A, lanes 6–7 and 11–13 and Fig. 5C). With increasing concentrations of APE1 up to 200 nM, the 16-mer cleavage products completely disappeared, suggesting complete degradation of the fragment by the 3′→5′ exonuclease activity of APE1 (Fig. 5A, lanes 8 and 14 and Fig. 5C). Curiously, excision of T next to dA-ALI was slightly higher compared to dA, leading to the possibility that the presence of a neighboring dA-ALI adduct provides easier access for TDGFL. Taken together, these results suggest that APE1 can remove bulky dA-ALI adducts at 3′-termini of DNA strand breaks with good efficiency. However, the TDG-catalyzed activity towards 3′ neighboring T next to dA-ALI is still very low to have physiological relevance even when combined with APE1 action.

Influence of bulky dA-ALI and II adducts on TDG-catalyzed futile excision of pyrimidines

The above results showed that after a long incubation time at 37 °C TDGFL is capable of excising T and C residues opposite to A and G in TpG and CpA contexts, respectively, in regular non-damaged DNA duplexes to a significant extent (Fig. 3A, lanes 7–9 and Figs. 4A, 4C, lanes 8 and 16). In addition, TDGFL can excise T opposite to dA-ALI adduct in an aberrant manner, but with much lower efficiency as compared to the futile excision of T in T•A base pairs at different positions of the same DNA duplex substrate (Fig. 3A). Since the results shown above and our recent published study (Manapkyzy et al., 2024) clearly demonstrated the absence of nuclease contamination in our enzyme preparations, we decided to use short blunt-end DNA duplexes for further characterization of TDG-catalysed repair. For this, 24 mer 14-07 and 10-05 duplex oligonucleotides with the same sequence, but containing either dA-ALI or dA-ALII adduct at position 14, respectively (Table 1), were incubated with TDGFL. It should be noted that the use of short DNA duplex substrates allowed us to reduce the number of cleavage sites as compared to long 55 mer oligonucleotide utilized to construct dmbDNA, which in turn simplified analysis and interpretation of the results. As shown in Fig. 6A, TDGFL exhibited efficient futile excision of thymine residues in the bottom and top strands of non-damaged regular A•T* and A*•T 10-05 duplex oligonucleotide after 1 or 18 h incubation (lanes 4–5 and 7–8). Intriguingly, the presence of dA-ALII adduct in dA-ALII•T* and dA-ALII*•T 10-05 duplexes dramatically reduced and changed the pattern of the futile activity (lanes 12–13 and 15–16). When acting upon regular A*•T 10-05 duplex, TDGFL excised in a futile manner 15% and 54% of T opposite to A at position 11 in the 32P-labelled top strand after 1 or 18 h, respectively, (lanes 7–8). In the 10-05 dA-ALII*•T duplex, excision of T11 was reduced approximately tenfold (lane 16), but the excision pattern remained the same, with T at positions 11 and 9 as preferred sites (lanes 7–8 vs 15–16). In a regular 10-05 A•T* duplex with the labelled bottom strand, TDG also excises T at position 11, but with ~5-fold less efficiency (Fig. 6A, lanes 4–5), as compared to the excision of T11 in the top strand of the same duplex (lanes 7–8). In addition, we observed a faint band migrating at position of 14 mer, suggesting that TDGFL excises C15 opposite to G, but with a very low efficiency (0.4%) (lane 5). Strikingly, the presence of dA-ALII adduct changed completely the pattern of futile activity on the bottom strand of 10-05 dA-ALII•T* duplex: TDGFL exhibits a 20-fold stimulation of excision of C15 (8.2%) opposite to G and a 7-fold inhibition of excision of T11 (1.4%) opposite to the dA-ALII adduct (lane 13) as compared to the regular A•T* 10-05 duplex (lane 5). It should be noted that we used 10-05 G•U15* duplex incubated with TDG to generate 14 mer size marker (lane 10), which migrates at the same position as the faint cleavage product in (lane 5) and major product in (lane 13). Next, we examined the pattern of TDGFL-catalyzed excisions of pyrimidines in 14-07 A•T and dA-ALI•T duplexes. As shown in Fig. 6C, the presence of dA-ALI adduct influences the futile removal of T and C by TDGFL in a manner very similar to that of dA-ALII (lanes 4–5, 7–8 and 10–11). Overall, these results suggest that: (i) TDG excises thymine opposite to dA-AL adducts poorly compared with its canonical G•T substrate; (ii) the presence of dA-AL adducts in duplex DNA rather inhibits futile excision of both opposite thymine in complementary strand and neighboring 5′ upstream thymine in the same adducted strand; (iii) depending on sequence contexts the presence of dA-AL adducts may also stimulate excision of neighboring regular cytosine in complementary strand.

To examine further the effect of sequence context and the presence of bulky purine adducts on TDG activities, we measured the DNA glycosylase activities using a 19-mer 14-06 duplex containing dA-ALI adduct at position 13 and a 24-mer 10-13 duplex with the same sequence as 10-05 but containing a dG-LII adduct at position 10 (Table 1). As shown in Fig. 7A, after 18 h of incubation TDGFL exhibited very weak futile activity towards short regular A•T* and A*•T 14-06 duplexes, since we observed excision of 3.1% of T11 opposite to A in the bottom strand and 2.8% of C8 opposite to G in the top strand of the A•T* 14-06 substrate (lanes 5 and 8). The presence of dA-ALI in 14-06 duplex did not change the pattern of futile activity, but stimulated excision of T11 opposite to A in the bottom strand of dA-ALI•T* threefold, while reducing the excision of C8 opposite to G in the top strand of dA-ALI*•T duplex also threefold (lanes 11 and 14).

As shown in Fig. 7C, similar to 10-05 duplex, TDGFL exhibited robust futile activities towards 24-mer regular A•T* and A*•T 10-13 duplexes, excising 22.5% of T11 in the bottom strand and 17.7% of T11 in the top strand (lanes 5 and 8). Replacement of G with the bulky dG-ALII adduct stimulated the excision of T11 in the bottom strand about threefold, whereas the excision of T15 opposite modified guanine remained very weak (lane 11). The excision of T11 immediately 3′ to dG-ALII in the top strand was also strongly inhibited (lane 14). No other changes in the pattern of futile activity were observed as compared with the regular 10-13 duplex. Altogether, the results suggest that (i) TDGFL inefficiently recognizes thymine opposite to either a bulky adenine or a guanine adduct compared with its canonical G•T substrate; (ii) the presence of a dG-ALII adduct depending on the sequence context, can strongly inhibit futile excision of immediately neighboring pyrimidines.

Discussion

Aristolochic acid (AA), an established human environmental carcinogen, is a mixture of several closely related plant compounds differing by the presence of one or two methoxy groups, and exerts its action through formation of adducts between its metabolically activated derivatives, aristolactams (AL), and exocyclic amino groups of adenine or guanine (Fig. 2, reviewed in (Sidorenko, 2020)). Mutations induced by exposure to AA both in clinical tumors and in cell culture are highly specific T→A transversions predominantly in the CpTpR (R = G or A) context and define the single base substitution signature SBS22 (Hoang et al., 2013; Poon et al., 2013). AL adducts in DNA are extremely long-living in vivo due to their slow repair (Grollman et al., 2007; Jelakovic et al., 2012; Sidorenko et al., 2012) thus presenting ample opportunities for mutations to arise and get fixed. However, the molecular events in the AA-induced mutagenesis following the adduct formation are still unclear.

Previously, we had demonstrated that human TDG can target a non-damaged DNA strand to remove regular T opposite adenine-derived lesions such as Hx or εA in a TpG/CpA* sequence context (where A* is a modified adenine residue). This aberrant removal of a normal base initiates the mutagenic BER cycle, which utilizes a damaged DNA template containing Hx or εA for repair synthesis, resulting in T→C or T→A mutations, respectively, with no need for DNA replication. If this mechanism is operational with other lesions, in particular bulky lesions such as dA-AL adducts, TDG could have a role in AA-induced mutagenesis in cancer cells. Studying the bypass of a dA-ALI adduct by using transfection of a gapped plasmid construct and TLS-deficient mammalian cell lines revealed that DNA polymerase ζ, one of the main human translesion DNA polymerase, can catalyze preferential insertion of dAMP opposite to dA-ALI adduct in certain DNA sequence contexts (Hashimoto et al., 2016).

There are examples of aberrant repair by other mismatch-specific DNA glycosylases that do not differentiate between damaged vs normal DNA strands when removing regular DNA bases, such as adenine-DNA glycosylases MutY in E. coli and MUTYH in human cells. Previously, it was demonstrated that bacterial MutY can excise regular A opposite a 8oxoG residue misincorporated during DNA replication, this mutagenic post-replicative repair resulting in higher rates of A→C transversions in wild-type and particularly in mutT E. coli strains (Fowler et al., 2003). It has been shown that human homolog of MutY, MUTYH can interrupt NER proteins during removal of UV lesions, possibly via aberrant excision of adenine residues in non-damaged DNA strand (Mazouzi et al., 2017). It should be stressed that the activity of MutY and MUTYH on an A•8oxoG base pair is only mutagenic if the 8oxoG is incorporated from an oxidized dNTP pool—there the removal of correct A in template strand could be mutagenic. Whereas, under normal conditions, a cell encounters rather an A•8oxoG base pair resulting from an A being inserted opposite an 8oxoG residue in template DNA strand—in this case, the excision of regular A by MutY and MUTYH is anti-mutagenic.

Based on these observations, we hypothesized that TDG, similarly to MutY, may initiate aberrant excision of thymine opposite dA-AL adducts, which in turn may contribute to characteristic mutations in cancer patients exposed to AA (Hoang et al., 2013; Poon et al., 2013). In the present study, we report that upon long incubation at 37 °C human TDG can excise T opposite to both regular A and bulky dA-ALI-II and dG-ALII adducts with low efficiency strongly dependent on the sequence context. Importantly, in control experiments we observed efficient TDG-catalyzed aberrant excision of mismatched T residue opposite to G and Hx residues in duplex DNA within the same sequence contexts. These biochemical results suggest that TDG hardly contributes to AA-induced mutagenesis in human cells, due to very low efficiency of excision and a lack of specificity towards T residues opposite to bulky aristolactam adducts when compared with T opposite to A in a regular duplex.

Mechanistically, our observations fit well into the known structures of TDG–DNA complexes (Coey et al., 2016; Maiti et al., 2008). To excise T or oxidized mC derivatives, the enzyme straddles the target DNA strand and inserts the β5/α6 loop into the minor groove to evert the substrate nucleotide into the active site pocket. A highly conserved Arg275 residue within this loop is especially important for maintaining this distorted DNA conformation through interactions of its bulky side chain and the positively charged guanidine moiety with the nucleobases and phosphates flanking the target nucleotide; replacement of this residue with Ala or Leu slows down the rate of T excision from G•T mispairs 8- and 30-fold, respectively (Maiti, Morgan & Drohat, 2009). Since dG-ALII significantly distorts the minor groove and presents a significant population of conformations where both AL and the opposite base are displaced towards the minor groove (Kathuria, Sharma & Wetmore, 2015), this lesion could interfere with productive enzyme binding, consistent with very poor T removal from opposite dG-ALII. On the other hand, dA-AL adducts displace the opposite base into the major groove (Kathuria et al., 2015; Kathuria, Sharma & Wetmore, 2015, 2016; Lukin et al., 2012) and do not encounter this problem. However, they still possess a strongly stacking AL system that sterically hinders the insertion of the “plugging” Arg275 side chain into the void vacated by the everted T (Figs. 8A, 8B), so the excision of T from opposite dA-ALI and dA-ALII is better than from mispairs with dG-ALII but rather inefficient in comparison with canonical substrate and even with the futile reaction. On the other hand, when TDG binds the lesion-containing strand in the register 3′ to the adduct, the interaction of Arg275 with the planar AL system might be favorable (Fig. 8C) and thus the enzyme would preferentially excise T from this position in a futile manner, as we observed in the 14-04 context (Fig. 4C, lane 16).

Figure 8 Scheme of interaction of TDG with substrate DNA.

Productive (A), aberrant (B) and futile mode (C) of interaction in the 14-04 DNA context. AL, dA-AL adduct. Small tribars symbolize interactions of the Arg side chain with the surrounding nucleobases (Maiti, Morgan & Drohat, 2009).

Conclusions

In the present work we further characterize the DNA substrate specificities of TDG, the well-known human mismatch-specific thymine-DNA glycosylase. The data demonstrate that TDG can excise T opposite to both regular A and bulky aristolactam-DNA adducts with high preference to TpG/CpA* context in duplex DNA. However, the low efficiency of this reaction means that AA-induced mutagenesis in humans most likely occurs independently of TDG. Based on these observations, we propose that the hot spots of mutagenesis observed in AA-induced cancers may be due rather to errors of translesion DNA synthesis than aberrant repair. Although we did not find solid evidence for the involvement of TDG in aberrant mutagenic repair of AA adducts, it cannot be excluded that post-mitotic cells, such as terminally differentiated neurons, or reversibly growth-arrested quiescent stem cells, may be prone to the aberrant and futile excision of canonical DNA bases in damaged DNA duplexes of another chemical nature. Additional studies are required to examine whether TDG—or other DNA glycosylases capable of excising canonical nucleobases—act either in an aberrant or futile manner on different DNA substrates.

Supplemental Information

Supplemental Information 1 Supplemental Tables.

Supplemental Information 2 Uncropped gel images used in Figure 3.

The action of TDGFL on dumbbell DNA duplexes containing dA-ALI.

Supplemental Information 3 Uncropped gel images used in Figure 4.

The action of TDGFL on different strands of dA-ALI•T* dmbDNA.

Supplemental Information 4 Uncropped gel image used in Figure 5.

Removal of bulky dA-ALI adduct in the top 27-mer strand of dA-ALI*•T dmbDNA through combined action of TDG FL and APE1.

Supplemental Information 5 Uncropped gel image used in Figure 6A.

The action of TDG FL on short blunt-end duplex oligonucleotides containing A, dA-ALI or dA-ALII.

Supplemental Information 6 Uncropped gel image used in Figure 6C.

The action of TDG FL on short blunt-end duplex oligonucleotides containing A, dA-ALI or dA-ALII.

The authors are indebted to the late Prof. Arthur P. Grollman for his role in initiating this work.

Abbreviations

TDG human native mismatch-specific thymine-DNA glycosylase

TDGFL native, full-length version of TDG

BER base excision repair

MBD4 methyl-binding protein 4

NER nucleotide excision repair

AA aristolochic acids

OGG1 human 8-oxoguanine DNA glycosylase

AP apurinic/apyrimidinic

APE1 major human AP endonuclease 1

Hx hypoxanthine

εA 1,N6-ethenoadenine

8oxoA 7,8-dihydro-8-oxoadenine

dA-ALI 7(deoxyadenosine-N6-yl)aristolactam I

dA-ALII 7-(deoxyadenosine-N6-yl)aristolactam II

dG-ALI 7-(deoxyadenosine-N2-yl)aristolactam I

Additional Information and Declarations

Competing Interests

The authors declare that they have no competing interests.

Author Contributions

Diana Manapkyzy performed the experiments, prepared figures and/or tables, authored or reviewed drafts of the article, and approved the final draft.

Gulzhan Zhamanbayeva performed the experiments, prepared figures and/or tables, and approved the final draft.

Viktoriya Sidorenko analyzed the data, authored or reviewed drafts of the article, designed and prepared DNA oligonucleotides containing bulky aristolactam-DNA adducts, and approved the final draft.

Radha Bonala performed the experiments, prepared figures and/or tables, designed and prepared DNA oligonucleotides containing bulky aristolactam-DNA adducts, and approved the final draft.

Francis Johnson conceived and designed the experiments, prepared figures and/or tables, authored or reviewed drafts of the article, designed and prepared DNA oligonucleotides containing bulky aristolactam-DNA adducts, and approved the final draft.

Bakhyt T. Matkarimov analyzed the data, prepared figures and/or tables, and approved the final draft.

Dmitry Zharkov conceived and designed the experiments, analyzed the data, prepared figures and/or tables, authored or reviewed drafts of the article, performed structural analysis, and approved the final draft.

Murat K. Saparbaev conceived and designed the experiments, analyzed the data, prepared figures and/or tables, authored or reviewed drafts of the article, and approved the final draft.

Sabira Taipakova conceived and designed the experiments, performed the experiments, analyzed the data, prepared figures and/or tables, authored or reviewed drafts of the article, and approved the final draft.

Data Availability

The following information was supplied regarding data availability:

The raw data is available in the Supplemental Figures and Tables.

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
