# Peer review of "Thymines opposite to bulky aristolactam-DNA adducts in duplex DNA are not targeted by human thymine-DNA glycosylase"

_PeerJ, doi:10.7717/peerj.19577_

## Round 0.1 · original submission · Major Revisions

· Academic Editor

Major Revisions

Please address the concerns of all the reviewers and amend the manuscript accordingly.

·

Basic reporting

The article is well written and uses professional English. The references are correct and appropriate. The authors have structured the article professionally, and the figures are clear. However, I was unable to download the raw data because the supplementary data link is associated with another article. The experiments performed are consistent with the experimental hypothesis.

Experimental design

The article raises an interesting scientific question in oncology within the journal's scope. The authors aim to establish whether TDG plays a role in urothelial cancer tumorigenesis induced by aristolochic acid (AA), leading to bulky aristolactam-dA (dA-AL) and aristolactam-dG (dG-AL) adducts. They designed glycosylase assays using duplex oligonucleotides containing T opposite to dA-AL and dG-A adducts within the CpA*/TpG context. The experimental procedures were performed correctly and rigorously, as reported in the materials and methods.

Validity of the findings

Although the authors hypothesize a novel role of TDG in tumorigenesis, they concluded it is unlikely TDG plays a role in AA-induced mutagenesis in human cells through aberrant repair mechanisms. They proposed that mutagenesis might result from inaccuracies in translesion DNA synthesis. The data demonstrated are solid and statistically significant.

Additional comments

The authors of this study investigated the hypothesis that TDG may play a role in the tumorigenesis of urothelial cancer induced by aristolochic acid (AA), leading to the formation of bulky aristolactam-dA (dA-AL) and aristolactam-dG (dG-AL) adducts. TDG may specifically facilitate A-T> T-A transversion mutations through aberrant excision of T opposite to dA-AL and dG-A adducts within the CpA*/TpG context. Nonetheless, glycosylase assays have demonstrated that human TDG excises T paired with regular A, as well as dA-ALI-II and dG-ALIIdA-ALI adducts, with limited efficiency. Consequently, the authors concluded that it is improbable that TDG plays a role in AA-induced mutagenesis in human cells through aberrant repair mechanisms. The authors proposed that mutagenesis might result from inaccuracies in translesion DNA synthesis. The article is well-composed, and the experimental design is robust. Therefore, I have only made minor comments.

1) The authors are encouraged to incorporate schematic illustrations within the figures to depict glycosyl assays. These illustrations should exemplify the expected fragment lengths generated based on the activity of TDG, as well as the specific sequences of the duplex oligonucleotides.
2) Have the authors tested TDG activity in oligonucleotides with T paired with -dA-AL or dG-AL in methylated or unmethylated CpG contexts? If so, what were the observations?

Reviewer 2 ·

Basic reporting

The manuscript is well written and professional. Literature references are very appropriate. Some of the figures will need minor additions as elaborated below. The study was hypothesis-driven.

Experimental design

The manuscript by Manapkyzy et al reports the results of a hypothesis-driven investigation which focuses on the potential activity of TDG on synthetic DNAs containing thymine bases opposite aristolactam-DNA adducts. The manuscript is well-written, and as with many investigations whose fundamental hypothesis is proven incorrect, there are an abundance of control experiments validating reagents and enzymes and these data provide important kinetic analyses concerning the activities of TDG on “undamaged” DNA substrates. The conclusion of the study is that TDG does not excise thymine when paired with either dA-AL or dG-ALII adducts and thus, does not modulate the mutagenicity associated with aristolochic acid exposures.
Points to be addressed:
The use of MutY as an example of “Aberrant repair” (lines 116-120 and lines 421-423) needs further clarification: as stated, the activity of MutY on an A:8-oxoG base pair is only mutagenic if the 8-oxoG is incorporated from a dNTP pool – there removal of the correct A is potentially mutagenic; the potentially more common occurrence of a cell encountering an A:8-oxoG base pair is when an A has been inserted opposite an 8-oxoGua lesion – in this case, the activity of MutY is anti-mutagenic.
As stated above, the experimental data are clearly presented and well controlled. Although the text indicates that all experiments were repeated at least 3 times (lines 240-241), the values reported in the text do not reflect the statistical analyses, such as standard deviations. This should be corrected throughout the Results.
Concerning data presentation in Figure 3: add red outlines on the gel images in Panels A and C that correspond to the values given at the bottom of Panels A and C;
Concerning data presentation in Figure 4: add red outlines on the gel images in Panel A that correspond to the values given at the bottom of Panel A; on line 328 change the apostrophe to a prime symbol; line 328, the text refers to APE1 reactions in lanes 4-6 – however, the image shown in lane 4 does not include APE1 – did the authors mean to say lanes 5-7? Also, in line 329, should lane 4 be lane 5?
Concerning data presentation in Figure 5: add red outlines on the gel images in Panels A and C that correspond to the values given at the bottom of Panels A and C; on line 361, the text refers to lanes 5-6 – should this be lanes 4-5? Line 363 mentions lane 5, should this be lane 4?
Concerning data presentation in Figure 6, add red outlines on the gel images in Panels A and C that correspond to the values given at the bottom of Panels A and C
If the above lane identifications are in fact errors, then the authors need to rigorously double-check their lane assignments throughout.



Minor points:
Line 57: “weak” should be changed to “inefficient” or “decreased” activity
Line 140: fix typo of “worls” to “world”

Validity of the findings

Although the overall finding of these studies has revealed a paucity of activity by TDG on multiple substrates, these data are still very valuable to the DNA repair and mutagenesis and carcinogenesis fields and should be published.

Reviewer 3 ·

Basic reporting

Clear and unambiguous, professional English used throughout.
English used in the manuscript is not always clear and unambiguous.
Some sentences are difficult to understand as it does not correlate clearly with the findings. For example, the figure legend of Fig.4 is “Figure 4. Removal of bulky dA-ALI adduct by TDGFL and APE1 from the top 27-mer strand of dA-ALI*●T dmbDNA.” From this sentence, it is not clear whether dA-ALI is removed by TDG or APE1. Also, in Figure 3A, the author mentioned that dA-ALI adduct does not cleave by TDG even after 18h incubation in CpXpG and ApXpT sequence context where X=dA-ALI. Instead of the dA-ALI adduct, the 3’ T is removed by TDG as in Figure 3C. The figure legend is very confusing. This confusion needs to be clarified.

Literature references, sufficient field background/context provided.
The introduction/ background is relatively long. I would suggest making it concise, describing the role of TDG in BER and aberrant and futile DNA repair activity. Also, a figure showing a schematic diagram of how TDG is involved in the BER mechanism with different substrate removal (aberrant and futile). It will help to understand readers not familiar with the TDG enzyme.

Professional article structure, figures, and tables. Raw data shared.
The article is written as instructed by the journal guidelines. Figures and tables are formatted correctly, but are difficult to follow. The raw data files present are of poor quality, also do not always match the figures presented in the manuscript.

Self-contained with relevant results to hypotheses.
No, the results and discussion do not relate directly to the hypotheses.

Experimental design

Original primary research within the Aims and Scope of the journal.
Yes, the manuscript is original research and falls within the scope of the journal.

Research question well defined, relevant & meaningful. It is stated how research fills an identified knowledge gap.
Yes, the authors define the gap in the literature. However, the use of too many oligonucleotides to address the question makes the manuscript somewhat complicated and sometimes difficult to follow.

Rigorous investigation performed to a high technical & ethical standard.
Ethics protocols were in place, and all samples were collected and analyzed according to common widely accepted methods.

Methods are described with sufficient detail & information to replicate.
Yes, the methods were described thoroughly. However, the only method used to analyze the data is gel electrophoresis of P32-labeled oligos, which depends on the quality of the gel. Sometimes it is difficult to measure if the gel quality is bad.

Validity of the findings

Impact and novelty not assessed. Meaningful replication encouraged where rationale & benefit to literature is clearly stated.
The rationale and benefit to the literature are clearly stated.

All underlying data have been provided; they are robust, statistically sound, & controlled.
The raw data files have been provided. However, the raw data files presented are of poor quality, also do not always match the figures presented in the manuscript.

Conclusions are well stated, linked to the original research question & limited to supporting results.
Yes, I have no concerns about the conclusions drawn from the data.

Additional comments

Based on previous findings (Talhaoui et al. 2014), the author hypothesized that TDG-initiated Base Excision Repair (BER) may be involved in generating mutations in the CpG-rich promoter region commonly observed in many cancer types. Based on their hypothesis, in this manuscript author investigated whether Thymine-DNA Glycosylase (TDG) participates or not in the aberrant removal of thymine opposite to bulky aristolactam-DNA adducts (a carcinogen) in duplex DNA, generated by aristolochic acid (AA). Long incubation of TDG at 37°C can excise thymine (T) and cytosine (C) from regular DNA and thymine opposite to dA-ALI, dA-ALII, or dG-ALII adducts in damaged DNA with low efficiency. The author concluded that human TDG is not involved in the aberrant DNA repair of AA-induced DNA damage due to its low efficiency.
comments-
1. In line 308, “---strand containing A or----” This A will be G.
2. In line 316, “---aduct at the 3’ terminus---”, this 3’ will be 5’
3. In line 332, “---3’-terminal---" this 3’ will be 5’
4. In line 338, “---at 3’-terminal of---" this 3’ will be 5’
5. Line 337-339, “taken together------T next to dA-ALI is quite low.” This is confusing.
6. Figure 4: The legend is very confusing and needs to be corrected. Also, the G●17T* marker used has not been described/explained well in the legend.
7. In Figure 5, the author used two blunt-end oligos, 10-05 and 14-07, despite the same sequence. And shows two gel figures for A-ALII (Fig. A) and A-LI (Fig. C) which is very confusing. Lanes 1 to 5 are the same for both figures. A and C. To make it simple, the author can show the findings in one gel, not in two separate gels. As the author differentiates the sequences denoting X=G, A, A-ALI, and A-ALII. It is confusing the reader using two oligos 10-05 and 14-07.
Also, this is not clear why the author used blunt-end DNA for this study. As the author mentioned in the Result section, lines 262 to 267, “In addition, to avoid nonspecific degradation----- we constructed a dumbbell-shaped duplex (dmbDNA)---oligonucleotide.”
The author needs to explain why they used blunt-end DNA instead of dumbbell-shaped duplex DNA. Is there any problem using dumbbell-shaped duplex DNA? The use of different DNA is some extent, confusing.
In Figure 5A, Lane 9-10, it is not clear how this band (cleavage product of G●U*, size 14mer) appeared from oligo 10-05. This needs to be clarified.
8. Line 370-373, author mentioned “Overall, these results suggest that: (i) TDG inefficiently excises thymine opposite to dA-AL adducts; (ii) the presence of dA-AL adducts in duplex DNA rather inhibits futile excision of neighboring thymine residues, but in certain sequence contexts may stimulate excision of cytosine residues.”
However, the results from Figure 5 do not agree with this statement. Actually, TDG shows low efficiency for T when opposite to dA-AL adducts.
9. Line 392-395, “Altogether, the results suggest that (i) TDGFL does not efficiently recognize thymine opposite to either a bulky adenine or a guanine adduct, (ii) the presence of a dG-ALII adduct depending on the sequence context, can strongly inhibit futile excision of immediately neighboring pyrimidines.” This statement again contradicts the previous statement of Lines 370-373.
10. Under the discussion section, lines 441-442, the author stated that “Mechanistically, our observations fit well into the known structures of TDG DNA 442 complexes (Coey et al. 2016; Maiti et al. 2008).” Instead, it is important to discuss in more detail the mechanism of the current findings using previous structural knowledge.
11. Also, line 443, what does “the β5/αF loop” mean? It might be a font issue. This needs to be fixed.
12. While discussing the “plugging” role of Arg275, it is important to cite the original paper (Maiti et al., JBC, 2009).
13. Figure 7 needs to be explained in more detail describing Arg275's role and citing the paper (Maiti et al., JBC, 2009)

---

## Round 0.2 · accepted · Accept

· Academic Editor

Accept

All issues pointed out by the reviewers were addressed, and the revised manuscript is acceptable now.

·

Basic reporting

As previously articulated in my earlier review, the article fulfills all the specified criteria, including proficient use of English, appropriate citation of literature, and the presentation of professional figures and results that substantiate the hypothesis.

Experimental design

The article aligns with the objectives of the journal, and the study is both original and conducted with methodological rigor.

Validity of the findings

The article contributes a novel insight into the function of TDG in the context of DNA repair.

Reviewer 2 ·

Basic reporting

The revised manuscript has been significantly revised, resulting in a significant improvement in the clarity of the presentation.

Experimental design

Excellent, with no additional suggestions.

Validity of the findings

Extremely well controlled and validated.

Additional comments

None

Reviewer 3 ·

Basic reporting

The authors employed clear and professional English throughout the manuscript. Sufficient background literature and citations have been provided. The article is professionally structured, incorporating the necessary figures, tables, and raw data.

Experimental design

The manuscript presents original research and falls within the scope of the journal. The authors clearly define the gap in the literature. Ethical protocols were followed, and all samples were collected and analyzed using widely accepted standard methods. The methods are described thoroughly.

Validity of the findings

The rationale and contribution to the literature are clearly stated. The raw data files have been provided. The conclusions are well articulated and appropriately linked to the original research question.

Additional comments

The authors have made a good-faith effort to address my concerns. The revised article and figures now present a clearer perspective. I recommend acceptance of the manuscript.